# The Psoriatic Arthritis 5-Thermometer Scales (PsA-5Ts): Measurement Properties of a New Multidimensional Composite Tool for the Quick Assessment of the Overall Health Status in Psoriatic Arthritis

**DOI:** 10.3390/jpm13071153

**Published:** 2023-07-18

**Authors:** Fausto Salaffi, Marina Carotti, Sonia Farah, Marco Di Carlo

**Affiliations:** 1Rheumatology Clinic, Ospedale “Carlo Urbani”, Università Politecnica delle Marche, 60035 Jesi, Italy; fausto.salaffi@gmail.com (F.S.); sonia.farah91@gmail.com (S.F.); 2Dipartimento di Scienze Cliniche Specialistiche e Odontostomatologiche, Università Politecnica delle Marche, Clinica di Radiologia, Azienda Ospedaliero–Universitaria delle Marche, 60126 Ancona, Italy; marina.carotti@gmail.com

**Keywords:** psoriatic arthritis, PsA-5Ts, patient-reported outcomes, disease activity indices

## Abstract

Background: Psoriatic arthritis (PsA) is a heterogeneous condition that is difficult to assess. The goal of this research was to evaluate the clinimetric properties of the Psoriatic Arthritis 5-Thermometer Scales (PsA-5Ts), a new patient-reported outcome (PRO) to measure the overall health status in PsA patients. Methods: The PsA-5Ts were compared to composite measures of disease activity (DAPSA, PASDAS, CPDAI) and PROs (PsAID-12 and SF-36). The convergent validity was assessed through the Spearman’s correlation coefficient and the discriminant validity through the receiver operating characteristic (ROC) curve analysis, applying the Minimal Disease Activity (MDA) as an external criterion. Results: The cross-sectional assessment included 155 patients. Significant high correlations were observed when comparing PsA-5Ts to composite indices of disease activity and PROs (all at significance levels of *p* < 0.0001). The PsA-5Ts subscales were highly significantly different in terms of MDA status (all at *p* < 0.0001). The PsA-5Ts had good discriminant validity like that of the DAPSA, CPDAI, PASDAS, and PsAID-12, and better than that of the SF-36, with an area under the ROC curve of 0.944 (65% CI 0.895–0.974). Conclusions: The PsA-5Ts are an easy-to-use PRO that can be integrated with disease activity indices in the assessment of PsA in daily clinical practice.

## 1. Introduction

PsA is a complex inflammatory condition affecting the peripheral or axial joints, which is commonly seen in individuals with psoriasis. It involves various inflammatory patterns, such as axial disease (sacroiliitis, spondylitis) and peripheral arthritis (typically affecting the lower limb joints in an asymmetrical and pauci-articular manner), as well as specific features such as enthesitis and dactylitis. PsA prevalence is estimated at 30.7% among psoriatic patients and 0.42% in the general population [1,2,3]. The combination of peripheral joint, axial, and skin involvement in PsA significantly impacts the patients’ functionality, well-being, and health-related quality of life (HRQoL) [4,5,6,7].

Different composite measures have been proposed for PsA [8,9,10,11,12,13,14]. The Composite Psoriatic Disease Activity Index (CPDAI) is a grid system introduced by the Group for Research in Psoriasis and Psoriatic Arthritis (GRAPPA), classifying the severity of peripheral arthritis, skin disease, spinal disease, enthesitis, and dactylitis [15]. The Disease Activity for Psoriatic Arthritis (DAPSA), derived from the Disease Activity in Reactive Arthritis (DAREA) composite measure, primarily focuses on the articular component of the disease [16,17]. More recently, the Psoriatic Arthritis Disease Activity Score (PASDAS) was developed using multiple linear regression and showed better discriminatory ability than other indices in distinguishing between high and low disease activity levels in PsA [18].

In addition to composite measures, the consideration of HRQoL has become increasingly important for individuals with PsA, influencing resource allocation, intervention design, and the use of biologic agents. Pain, fatigue, physical function, skin problems, and emotional well-being hold greater relevance and meaning for patients than composite disease activity measurements [6,18,19,20]. Recognizing the significance of patient perspectives, organizations such as the Outcome Measures in Rheumatology Clinical Trials (OMERACT) and the American College of Rheumatology (ACR) emphasize the assessment of patient-reported outcomes (PROs) in clinical trials and in daily practice [19,21,22,23,24,25]. PROs allow for the evaluation of symptoms, functions, and other aspects of life negatively influenced by the disease. Pain, skin problems, fatigue, and resulting physical limitations also impact social functioning and mental health, further diminishing the quality of life for patients with PsA [6]. Research has shown that patients with comorbid depression and PsA experience worse long-term outcomes, higher comorbidity rates [26], and increased mortality rates [27]. PRO instruments have been extensively used worldwide and demonstrated strong associations with objective measures in rheumatologic disorders as well as other chronic conditions such as cancer, asthma, hypertension, heart disease, stroke, mental illness, migraines, and diabetes [23,24,28].

Despite the availability of various PRO measures and the theoretical literature surrounding them, there is currently no standardized approach for their application in clinical practice, and a consensus on their optimal utilization is lacking. This lack of standardization limits the usefulness of clinical trial evidence in informing healthcare decisions, and the administration, scoring, and interpretation of PROs can be challenging in clinical practice. The graphical presentation of each instrument also significantly impacts its psychometric properties. For adults, numerical rating scales (NRS) and verbal descriptor scales (VDS) are typically preferred, as they may have difficulties with other types of scales [29]. Thermometer scales, which combine a modified vertical VDS with a graphic thermometer, have also been validated as pain measures in older adults and are recommended and widely used in clinical practice for inflammatory arthritis [29].

Based on these considerations, the aim of this study is to validate a new PRO to be used in patients with PsA, called the Psoriatic Arthritis-5 Thermometer Scales (PsA-5Ts), precisely involving five “thermometers” integrating NRS and VDS.

## 2. Materials and Methods

### 2.1. Patients Characteristics

This cross-sectional study included patients aged 18 and above who had been diagnosed with PsA based on the Classification Criteria for Psoriatic Arthritis (CASPAR) group criteria [30] or the Assessment of SpondyloArthritis International Society (ASAS) classification criteria [31,32] for those with axial involvement. Inflammatory markers, IgM rheumatoid factor levels, the presence of typical PsA periosteal new bone formation on radiographs, and evidence of axial involvement diagnosed by radiographs or MRI were reported based on agreement between the radiologists and referring rheumatologist. The patients with PsA were categorized into subgroups according to their cumulative disease pattern: monoarthritis, oligoarthritis (involving 2 to 4 joints), polyarthritis (involving 5 or more joints), and psoriatic spondylitis [33]. The presence of arthritis mutilans, characterized by shortened fingers with extensive skin folds, hypermobile joints, and stretched digits, was also noted. The radiographs were evaluated for sacroiliitis using the modified New York criteria [34]. The anatomical area of the axial skeleton to be investigated by MRI, including the sacroiliac joints, was determined by agreement between the rheumatologist and radiologist based on the patient’s complaints [32,35]. The choice of pharmacological treatment was determined by the managing physician based on their clinical judgment, since this study was not randomized [36]. Patients with coexisting fibromyalgia, symptomatic osteoarthritis, crystal-induced arthritis, or with other rheumatologic conditions interferring with the clinimetric assessment were excluded from the study. The other exclusion criteria included active skin diseases other than psoriasis, a history of cancer or lymphoproliferative disease, uncontrolled diabetes, unstable ischemic heart disease, congestive heart failure, active inflammatory bowel disease, chronic leg ulcers, recent stroke, positive hepatitis B serology, positive human immunodeficiency virus status, dementias, or other neurological diseases.

All patients included in the study were attending the outpatient and inpatient clinics of the Rheumatology Clinic of the Università Politecnica delle Marche (Jesi, Ancona, Italy). They were considered a representative sample of PsA patients referred to the department in a real-life setting. 

### 2.2. Variables

The patients received a comprehensive questionnaire package comprising socio-demographic information, quality of life questions, and disease-related factors. The socio-demographic factors encompassed age, gender, and educational level, while disease-related factors consisted of the disease duration (years since PsA diagnosis), comorbidites, and scores utilized to evaluate disease activity. The selection of domains and measures was based on their performance in previous research and their endorsement by GRAPPA members. These measures have been recognized as crucial components of psoriatic disease documentation at various OMERACT conferences [11,12,13,20,21].

The assessment encompassed various aspects including a peripheral joint evaluation (68 joints for tenderness (68 TJC); 66 joints for swelling (66 SJC)), patient-reported pain measured on an 11-point NRS, physician and patient assessments of disease activity (PhGA, PtGA) on an 11-point NRS, the patient’s general health status (GH) on a 0–100 NRS, dactylitis, and enthesitis evaluated with the Leeds Enthesitis Index (LEI). The LEI includes an assessment of tenderness at six specific sites (bilateral Achilles’ tendon insertions, medial femoral condyles, and lateral epicondyles of the humerus). The tenderness was evaluated dichotomously at each location, with 0 indicating not tender and 1 indicating tender [8]. 

The severity of skin involvement was assessed with the Psoriasis Area Severity Index (PASI). It considers the area of skin involvement, erythema, plaque thickness, and degree of scaling. The PASI score ranges from 0 to 72 in increments of 0.1 units, with higher values indicating more severe psoriasis [8].

The functional status was evaluated with the Health Assessment Questionnaire (HAQ). It comprises 20 questions that assess dressing and grooming, rising, eating, walking, hygiene, reach, grip, and other common daily tasks. The patients rate the difficulty of each activity on a scale of 0 to 4, with 0 representing the easiest and 4 indicating the most challenging [37]. The total scores ranges from 0 to 3, with higher scores indicating greater impairment.

#### 2.2.1. Minimal Disease Activity (MDA) Criteria

The Minimal Disease Activity (MDA) is a set of criteria used to determine the “level of disease activity judged a useful therapy target by both the patient and the physician, given existing treatment options and restrictions”. The MDA is defined as a patient fulfilling 5 out of 7 criteria: 68 TJC ≤ 1; 66 SJC ≤ 1; PASI ≤ 1; NRS-pain ≤ 1.5; PtGA of disease activity ≤ 2; HAQ ≤ 0.5; tender entheseal sites ≤ 1 [38].

#### 2.2.2. Disease Activity for Psoriatic Arthritis (DAPSA)

The DAPSA is the algebraic sum of 68 TJC, the 66 SJC, NRS pain, PtGA, and C-reactive protein (CRP) levels in mg/dl. The recommend cut-off values are ≤4 for remission (REM), >4 and ≤14 for low disease activity (LDA), >14 and ≤28 for moderate disease activity (MDA), and >28 for high disease activity (HDA) [39].

#### 2.2.3. Composite Psoriatic Disease Activity Index (CPDAI)

The CPDAI is a domain-based measure of PsA disease activity, including peripheral arthritis, skin, enthesitis, dactylitis, and spinal disease. The latter, if present, is evaluated with the Bath Ankylosing Spondylitis Disease Activity Index (BASDAI) [40,41]. The domains are ranked from 0 to 3, with empirical disease severity and activity cutoffs suggested in each of them. The sum of each domain scores yields overall composite scores ranging from 0 to 15 [15].

#### 2.2.4. Psoriatic Arthritis Disease Activity Score (PASDAS)

The PASDAS includes PhGA, PtGA, the Short-Form 36 Health Survey Questionnaire (SF-36) physical component scale (PCS), 68 TJC, 66 SJC, the LEI, a tender dactylitis count, and CRP levels. 

The PASDAS was calculated as: (0.18 √ physician global VAS + 0.159 √ patient global VAS – 0.253 √ SF-12 PCS + 0.101 LN (SJC + 1) + 0.048 LN (TJC + 1) + 0.23 LN (LEI + 1) + 0.377 LN (dactylitis count + 1) + 0.102 ln (CRP + 1) + 2) × 1.5 [18].

#### 2.2.5. Psoriatic Arthritis Impact of Disease 12-Item (PsAID-12)

The PsAID-12, developed by the European League Against Rheumatism (EULAR), has been translated and validated for use in PsA and is freely available [42,43]. Its compilation is quick and straightforward, making it feasible and widely applicable. The questionnaire consists of 12 domains that patients perceive as important for their health, with each domain rated on 0–10 NRS with different weights. The final PsAID-12 score is calculated as follows: (PsAID1 − pain NRS value (range 0–10) × 3) + (PsAID2 − fatigue NRS value (range 0–10) × 2) + (PsAID3 − skin NRS value (range 0–10) × 2) + (PsAID4 − work and/or leisure activities NRS value (range 0–10) × 2) + (PsAID5 − function NRS value (range 0–10) × 2) + (PsAID6 − discomfort NRS value (range 0–10) × 2) + (PsAID7 − sleep NRS value (range 0–10) × 2) + (PsAID8 − coping NRS value (range 0–10) × 1) + (PsAID9 − anxiety NRS value (range 0–10) × 1) + (PsAID10 − embarrassment NRS value (range 0–10) × 1) + (PsAID11 − social life NRS value (range 0–10) × 1) + (PsAID12 − depression NRS value (range 0–10) × 1). The total is divided by 20 to obtain total scores ranging from 0 to 10, with 10 representing the worst health. The proposed cut-off values are ≤1.4 for REM, >1.4 and ≤4.1 for LDA, >4.1 and ≤6.7 for MDA, and >6.7 for HDA [43].

#### 2.2.6. Short-Form 36 Health Survey Questionnaire (SF-36)

The SF-36 is a generic questionnaire used to assess health in various contexts [44,45]. The SF-36 consists of items grouped into eight scales (physical functioning, role limitations due to physical function, bodily pain, general health, mental health, role limitations due to emotional health, social functioning, and vitality). The raw domain scores are scaled from 0 to 100, with higher scores indicating better health. These scores are further transformed and weighted to generate PCS and Mental Component Summary (MCS) scores [44]. The SF-36 has demonstrated reliability, validity, and sensitivity to change [44,45,46,47].

### 2.3. The Psoriatic Arthritis 5-Thermometer Scales, Development and Description

In the development of the self-administered instrument for assessing disease activity, the content pool generation and item reduction and validation analysis are important initial steps. The most critical phase is item generation, as the final instrument can only include items identified in this stage, making its accuracy crucial.

#### 2.3.1. Item Pool Development

To create the preliminary questionnaire, a Delphi approach was employed to select questions [48]. This involved a comprehensive technique that had been previously used [49,50]. First, a literature review was conducted to select relevant items related to PsA disease assessment, followed by item creation. Predefined areas of screening were culled from four existing screening questionnaires, either generic and disease-specific, namely the SF-36, HAQ, PsAID-12, and BASDAI, resulting in the identification of 17 domains. The literature search with the definition of the 17 items was carried out by the authors. The patients were not involved in the item pool generation. Then, the Delphi method was applied, involving 22 rheumatologists experienced in the diagnosis and treatment of PsA, who were asked to establish an order of importance within the 17 health domains identified. 

The experts ranked the relevance of each domain in determining the severity of PsA using a Likert scale from 0 to 3, where 0 indicated unimportance and 3 indicated high significance. The average relevance scores were calculated for each domain, and a mean score of at least 2.0 (on a scale of 0 to 3.0) was required for a domain to be considered for inclusion in the questionnaire. The frequency at which each domain was rated by the 22 physicians was also considered, using Lynn’s process for content validation [51]. This is the quantitative phase, which measures the proportion or percent of experts who agree about the relevance of the items. The Content Validity Index (CVI) is used to establish proportion or percent of agreement among the experts. Lynn recommended the use of a relevance rating scale providing ordinal-level data through four Likert-like choices (4 = extremely relevant, extremely important; 3 = very relevant, very important; 2 = somewhat relevant, somewhat important; 1 = irrelevant, unimportant). Only the proportion of items receiving ratings of 2, 3, and 4 constitute the actual CVI, and any items rated at level 1 should be eliminated. The CVI formula is represented as follows: CVI or % agreement = number of experts agreeing on items rated as 2, 3, or 4/total number of experts. The items were considered to have adequate content validity if they achieved an agreement rate of 88% or higher. The questionable items ranged from 70 to 88% agreement, and the items were found to have unacceptable content validity if they achieved an agreement of 69% or lower.

Table 1 provides the CVI, mean importance, and FIP values for individual items. A final five-item model (pain, fatigue, physical function, skin problems, and depression), receiving adequate validity scores ranging from 71 to 98% agreement, remained in the assessment tool. The remaining 12 items achieved a rating of below 69% agreement, signifying unacceptable validity. These items were eliminated without further considerations (Table 1). Finally, the final five factors were weighted using a calculation formula similar to the approach used for the PsAID-12 questionnaire (18) to generate a 0–100 possible score: 5T-PROs = (5T-PROs1 − pain numeric thermometer value (range 0–10) × 3) + (5T-PROs2 − fatigue numeric thermometer value (range 0–10) × 2) + (5T-PROs3 − physical function numeric thermometer value (range 0–10) × 2) + (5T-PROs4 − skin problems numeric thermometer value (range 0–10) × 2) + (5T-PROs5 − depression numeric thermometer value (range 0–10) × 1) (Figure 1). The Italian language translation of the five questions was performed with consensus of the authors.

#### 2.3.2. Testing the Provisional Questionnaire

Pre-testing of the PsA-5Ts questionnaire was conducted to ensure that the wording was clear and the patients interpreted the items as they were intended. The questionnaire was administered to a group of 29 patients suffering from PsA, who were not included in the final study. To examine the participants’ level of comprehension of the instruments’ contents, a proxy question was asked: “Did you have any difficulty understanding the questionnaire items?” (to be answered on a five-point Likert scale).

### 2.4. Statistical Analysis

In the study, descriptive statistics were computed to summarize the data. For continuous variables, such as scores, means with standard deviations (SDs) or medians with 95% confidence intervals (95% CIs) were presented, depending on the distribution of the data. The Shapiro–Wilk test was used to determine if the data followed a normal distribution.

The categorical data were presented as proportions. To compare the demographic and clinical measures, the Mann–Whitney U test or Kruskal–Wallis test was used for continuous variables, while a chi-square analysis was used for categorical variables. Statistical significance was defined as *p*-values < 0.05.

Correlations between variables were analyzed using Spearman’s coefficient of rank correlation (rho). The strength of the correlations was interpreted based on the magnitude of the coefficient, with values > 0.90 considered very high, 0.70–0.89 as high, 0.50–0.69 as moderate, 0.26–0.49 as low, and ≤0.25 as little to no correlation.

To evaluate the discriminative accuracy of the PsA-5Ts scores in determining active and non-active disease states, a receiver operating characteristic (ROC) curve analysis was performed. The MDA criteria were used as an external criterion. The ROC curves were created by plotting the sensitivity against 100 – specificity for various cut-off points. The area under the ROC curve (AUC) was calculated to quantify the discriminative accuracy. AUC values ranging from 0.50 to about 0.70 represent poor accuracy, values from 0.70 to 0.90 are considered useful for some purposes, and higher values indicate high accuracy. The non-parametric Wilcoxon’s signed ranks test was used for calculation and comparison of the AUCs.

All data were entered into a Microsoft Access database designed for cross-sectional multicenter management. The analysis was performed using MedCalc^®^ version 17.0 (MedCalc Software, Mariakerke, Belgium).

## 3. Results

### 3.1. Testing the Provisional Questionnaire

Preliminary testing of the PsA-5Ts was conducted on 29 PsA patients (19 females and 10 males), aged from 26 to 73 years (mean 48.9 ± 11.8 years). Most of the patients (93.1%) found the items understandable. Two respondents found ‘some difficulty’ in understanding and responding to the items. After this provisional testing, no changes were made to the questionnaire.

### 3.2. Patient Characteristics

The study included 155 PsA patients (92 women and 63 males). The most prevalent pattern was represented by oligoarticular involvement (53.5%). Isolated DIP recruitment was seen in twenty-six individuals (16.8%) with peripheral arthritis. Only 6.4% of our patients had an isolated spondylitis, whereas 48.4% had at least one enthesis involved. The mean age was 56.81 ± 11.36 years, with a range of 19 to 78 years. The mean PsA duration was 8.36 ± 5.23 years. Most of the patients had several comorbid conditions, with a median of three (range from 1 to 4). One hundred and thirty-five patients with PsA (87.1%) were using disease-modifying antirheumatic medications or biological agents, and 50 patients (32.2%) were also receiving low-dose corticosteroids. The descriptive statistics of all clinical variables, composite disease activity indices, and HRQoL scores are shown in Table 2.

### 3.3. Descriptive Statistics of Composite Disease Activity Indices and Patient Self-Report Questionnaires

Table 3 provides the descriptive data for three composite disease activity indices (DAPSA, CPDAI, and PASDAS), each of which has a different weighting factor. The composite scores were not normally distributed (as shown by the Shapiro–Wilk test for normal distribution), and the distribution in all subjects was of the bimodal type, which was most likely relating to the different types of cases enrolled. Among the medians, the DAPSA was 25.00 (95% CI 20.27–27.15), the CPDAI was 8.00 (95% CI 7.00–9.00), and the PASDAS was 5.00 (95% CI 4.45–5.23).

The PsA-5Ts values were non-normally distributed (tested with the Shapiro–Wilk test for normal distribution) (Figure 2), as were the other composite disease activity indices and patient self-report questionnaires, with a median value of 42.50 (95% CI 38.00–45.26).

### 3.4. Convergent Validity

Significant high correlations were observed when comparing the PsA-5Ts to continuous composite indices of disease activity and PROs (all at significance levels of *p* < 0.0001) (Table 4). Considering the composite disease activity indices, the highest correlation coefficient was found between the PsA-5Ts and PASDAS (rho = 0.694). Correlations were identified between the DAPSA and CPDAI, which were statistically significant but less robust (rho = 0.664 and rho = 0.582, respectively). A correlation coefficient of 0.735 was found between the PsA-5Ts and PsAID-12, the other disease-specific PRO.

### 3.5. Discriminant Validity

Thirty-four (21.9%) patients met the MDA criteria. In these patients the mean PsA-5Ts score was 13.97 versus 45.53 in the patients not meeting the MDA criteria (*p* < 0.0001). The PsA-5Ts subscales differed substantially in patients meeting or not meeting the MDA criteria (Kruskal–Wallis test, all scores at *p* < 0.0001) (Figure 3).

The AUC-ROC curves for distinguishing patients meeting or not meeting the MDA were similar for all indices (Figure 4), with the best being that of the PASDAS and the worst being that of SF-36. The PsA-5Ts had excellent discriminating power, with an AUC of 0.944 (95% CI 0.895–0.974) (Table 5).

## 4. Discussion

This paper describes the validation of a new PRO, the PsA-5Ts, to assess the overall health status in a condition of complex clinimetry such as PsA. Alongside traditional measures of joint, enthesitic, axial, and cutaneous inflammation, the perspectives on PsA clinimetry in recent years have moved in favor of measurements increasingly focused on the patient’s subjective experience and how it impacts their quality of life. Being able to assess these aspects also means making a personalized choice of treatment and evaluating its effectiveness [51].

The objective of the PsA treatment has shifted towards achieving remission or tighter management thanks to the availability of biological therapies and aggressive disease-modifying anti-rheumatic drugs. This treatment approach aligns with the treat-to-target method (T2T), advocated by the T2T international task force [52] and the GRAPPA/OMERACT group, which proposes the MDA criteria as treatment targets [21]. Unlike other metrics, the MDA is applicable to all PsA patients, regardless of their disease pattern [53], making it suitable for individuals with polyarticular and oligoarticular arthritis.

In recent years, there has been a growing emphasis on a patient-centered approach to disease management, incorporating PROs. PROs are measures that capture patients’ own perceptions of their health and disease. Within academic research, the number of scientific articles regarding PROs has grown tremendously, highlighting the scientific community’s interest in putting the patient at the center. However, although the topic has undergone accelerated evolution in the last period, some tools have been available for more than 30 years. One of the main problems is the correct choice of instrument so that it can capture what it is intended to measure [54]. The creation of a new PRO must start with a conceptual model on what is to be measured, defining the boundaries well. From there, the implementation of the items starts, starting with expert opinions possibly supplemented with input from the patients. The validity of the new instrument must then be tested on a large scale [55].

PROs have become integral in assessing PsA patients, allowing for the inclusion of individuals with active domains that may be overlooked by measures focusing on multiple disease areas [11,20,21,38]. The PsAID-12, comprising physical and psychological dimensions, has been proposed as a comprehensive tool for assessing PROs in PsA patients [42,43]. Pain, fatigue, and skin problems emerge as crucial factors in this index [56]. While shorter questionnaires are preferred in routine clinical practice due to their ease of administration and interpretation, visual aids such as verbal descriptors within the NRS are commonly used, particularly for older adults [57,58].

The PsA-5Ts, a simple multidimensional tool incorporating NRS and VDS, effectively captures the five key areas of disease impact as perceived by physicians. It encompasses all core outcome measures recommended for use in randomized clinical trials for PsA [8,11,12,13,19,21]. Pain, the primary symptom in PsA, significantly affects impairment and is included in the PsA core set [59,60]. Fatigue, a prevalent symptom characterized by physical and mental decline, is associated with inflammatory conditions, chronic pain, reduced physical fitness, sleep disturbances, diminished quality of life, and emotional disorders [61,62]. Physical function, which plays a crucial role in illness treatment, encompasses muscular strength, endurance, range of motion, and the ability to carry out daily tasks. It is a core domain assessed in all PsA clinical trials [60,63]. Psychiatric comorbidities, particularly depression, are common in PsA and have adverse consequences [26,27]. Skin involvement in PsA affects the overall disease activity and impacts patients’ healthcare utilization, mirroring the well-documented psychological and social consequences seen in psoriasis [64,65,66]. The PsA-5Ts, by grouping the most relevant health domains in patients with PsA into five, goes some way toward that clinimetric simplification, which is most desirable, especially in complex-to-measure conditions such as seronegative spondyloarthritis [67].

It is important to note some limitations of this study. One limitation is the lack of evidence regarding the responsiveness to change or other psychometric tests. Additionally, the absence of an established “gold standard” instrument for assessing the multidimensional impact of PsA limits the ability to determine the validity of the criteria. However, this research provides a well-structured and well-executed validation of the PsA-5Ts in a large group of PsA patients. The findings may not be generalizable to PsA patients treated by general practitioners or in smaller practices since the participants were drawn from a tertiary center. Lastly, variations in recall time among the different measurements are common with self-report measures but should be considered when interpreting the results.

In conclusion, this study represents an initial step in establishing the psychometric properties of the PsA-5Ts. The questionnaire’s brevity, ease of interpretation, and non-stigmatizing format are the advantages of this instrument. It addresses critical aspects of PsA that are often overlooked as primary endpoints in research and clinical practice. It facilitates communication between patients and healthcare providers on a broader range of issues, as well as the identification of relevant services and resources. However, further investigation is necessary to assess its responsiveness to change. Future PRO collection methods could benefit from visualization techniques to enhance their comprehension and engagement among patients.

## Figures and Tables

**Figure 1 jpm-13-01153-f001:**
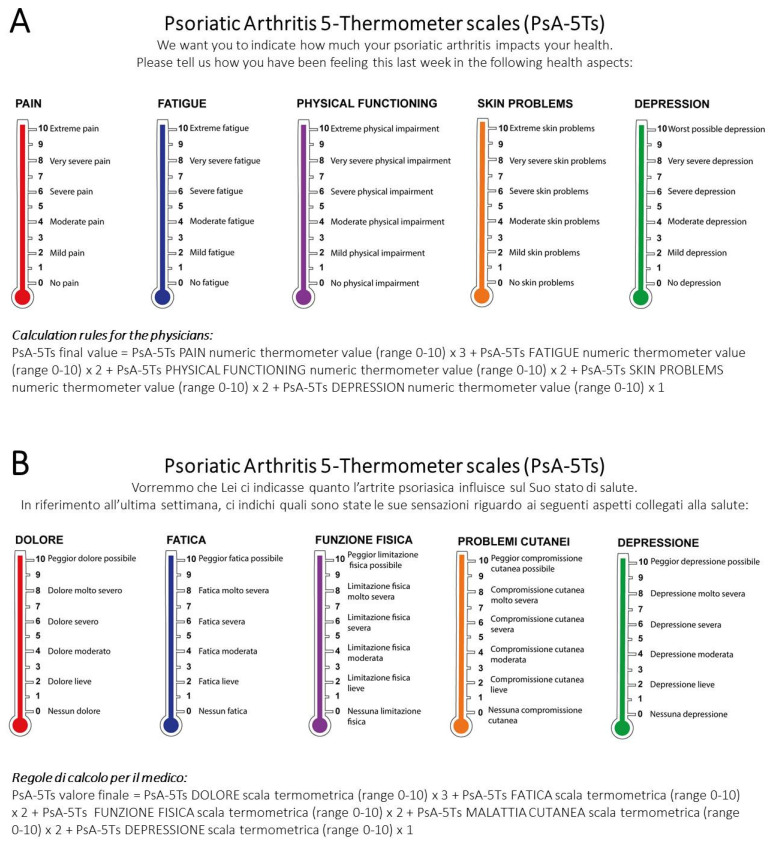
The PsA-5Ts sheet and calculation rules, English (**A**) and Italian (**B**) versions.

**Figure 2 jpm-13-01153-f002:**
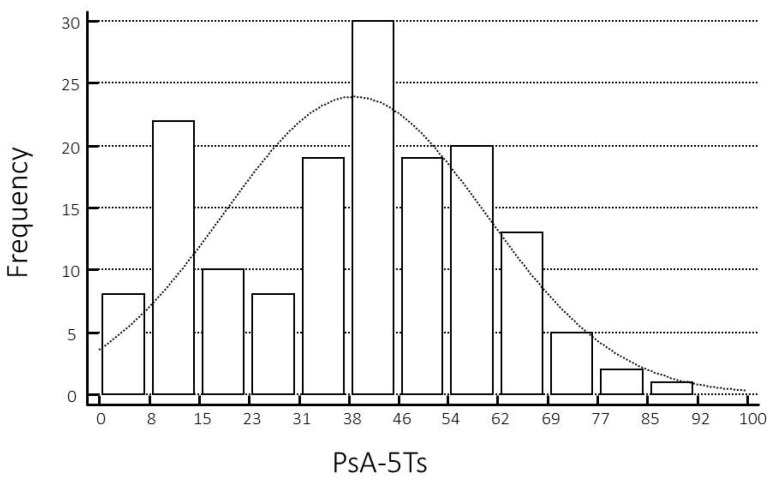
Distribution of PsA-5Ts scores. Columns represent the frequency of each score, dotted line represents the theoretical normal distribution.

**Figure 3 jpm-13-01153-f003:**
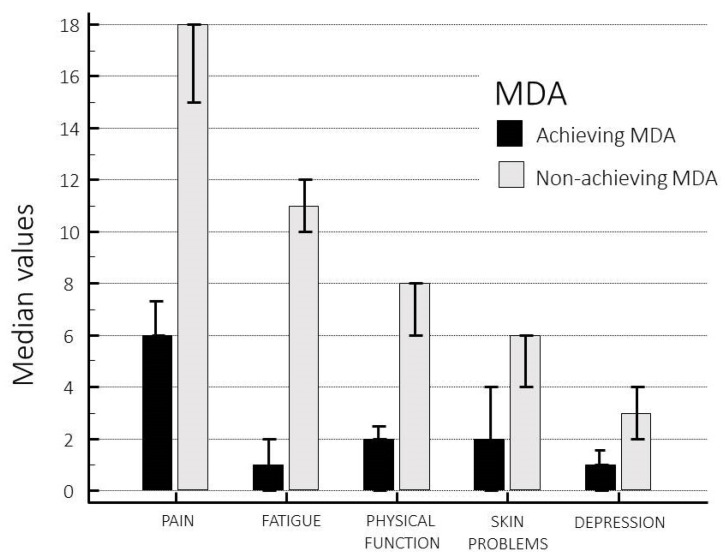
PsA-5Ts subscores according to MDA status (median and 95% CI for the median).

**Figure 4 jpm-13-01153-f004:**
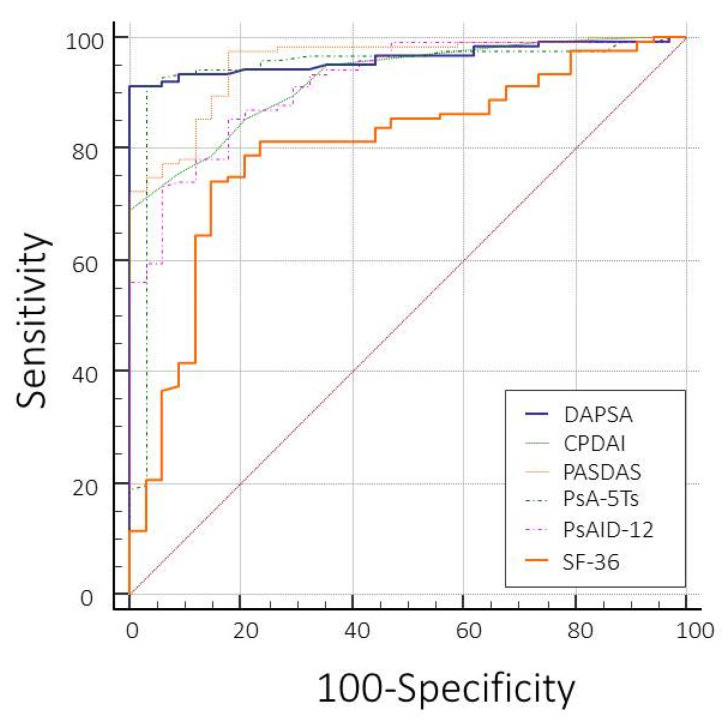
The ROC curve analysis (AUC-ROC curves values, standard errors, and 95% confidence intervals) for the discriminatory power of the disease activity, using the MDA as an external criterion, according to the different composite disease activity indices (DAPSA, CPDAI, and PASDAS) and patient-reported outcomes (PsA-5Ts, PsAID-12, and SF-36).

**Table 1 jpm-13-01153-t001:** Seventeen domains of health identified as important by 22 rheumatologists and their Content Validity Index, mean importance, and Frequency Importance Product scores.

Domain Number (Ordered According to FIP)	Domain and Short Defining Statement	% of Rheumatologists Considering This Domain a Priority (CVI)	Mean Importance	FIP
1	Pain (pain in joints, spine, and skin) *	98%	2.93	287.14
2	Fatigue (being physically tired but also mental fatigue, lack of energy) *	90%	2.16	194.40
3	Functional capacity (capacity to perform daily physical activities, loss of independence) *	88%	2.18	193.60
4	Skin problems (including itching) *	87%	2.20	191.40
5	Depressive mood (feeling sad or depressed) *	71%	2.01	142.71
6	Patient global assessment (PtGA) of disease activity (patient’s overall experience of their disease)	68%	1.87	127.16
7	Ability to work and for leisure (ability to work and/or do leisure activities)	63%	1.83	115.29
8	Sleep disturbance (sleep quality, sleep interruptions)	61%	1.82	111.02
9	Feeling of discomfort (discomfort and annoyance with everyday tasks)	56%	1.60	89.60
10	Anxiety, fear, and uncertainty (e.g., about the future, treatments, fear of loneliness)	55%	1.61	88.55
11	Embarrassment or shame due to appearance (feeling embarrassed/ashamed due to appearance)	52%	1.65	85.80
12	Social participation (participating fully in social activities)	49%	1.69	82.81
13	Relationship with family (relationship with family or people very close to you)	47%	1.60	75.20
14	Concentration difficulties (difficulty concentrating and memorising)	45%	1.58	71.10
15	Sexual life (sexual difficulties or dissatisfaction)	40%	1.80	72.00
16	Coping (adjustment to the disease, managing, being in charge, making do with the disease)	37%	1.61	63.27
17	Financial impact (experiencing financial loss due to treatment cost, work loss, early retirement, cost of assistive devices, etc)	35%	1.70	59.50

Abbreviations and legend: CVI = Content Validity Index; FIP = Frequency Importance Product; * = retained items.

**Table 2 jpm-13-01153-t002:** Demographic and clinical characteristics of the study population and of scores for each scale (n = 155).

Variables	Mean	Median	SD	IQR
Age (years)	56.81	56.00	11.36	49.00–65.25
Disease duration (years)	8.36	8.00	5.23	3.00–12.00
Educational level (years)	11.13	13.00	3.25	8.00–13.00
Comorbidites (number)	2.61	3.00	1.91	1.00–4.00
Sangha Index	4.79	4.65	3.60	1.55–7.02
TJC (68 joints)	6.73	6.00	6.66	0.00–11.00
SJC (66 ioints)	3.71	4.00	3.87	0.00–6.00
CRP (mg/dL)	3.62	3.11	2.95	1.63–4.75
LEI	1.41	1.00	3.38	0.00–2.00
Dactilitis (count)	2.19	1.00	2.42	0.00–4.00
PASI	5.41	4.40	5.09	1.17–8.52
HAQ	1.08	1.12	0.70	0.50–1.62
PsA-5Ts	39.39	42.50	20.18	16.00–59.60
DAPSA	23.34	25.00	12.98	9.67–32.67
CPDAI	7.89	8.00	4.48	4.00–11.25
PASDAS	4.46	5.04	1.74	2.99–5.70
PsAID-12	4.90	3.50	4.11	3.74–4.59
SF-36	41.14	39.19	9.55	37.20–41.62

Abbreviations: SD = standard deviation; IQR = interquartile range; TJC = tender joint count; SJC = swollen joint count; CRP = C-reactive protein; LEI = Leeds Enthesitis Index; PASI = Psoriasis Area Severity Index; HAQ = Health Assessment Questionnaire; PsA-5Ts = Psoriatic Arthritis-5 Thermometer Scales; DAPSA = Disease Activity for Psoriatic Arthritis; CPDAI = Composite Psoriatic Disease Activity Index; PASDAS = Psoriatic Arthritis Disease Activity Score; PsAID-12 = Psoriatic Arthritis Impact of Disease 12-item; SF-36 = Medical Outcomes Study Short-Form 36.

**Table 3 jpm-13-01153-t003:** Descriptive statistics of composite disease activity indices.

	DAPSA	CPDAI	PASDAS
Lowest value	3.92	0.00	0.54
Highest value	58.20	19.00	8.33
Arithmetic mean	23.34	7.76	4.49
95% CI for the arithmetic mean	21.29–25.38	7.07–8.45	4.22–4.75
Median	25.00	8.00	5.00
95% CI for the median	20.27–27.15	7.00–9.00	4.45–5.23
Variance	168.68	19.27	2.87
Standard deviation	12.98	4.38	1.69
Relative standard deviation	0.55 (55.64%)	0.56 (56.54%)	0.37 (37.77%)
Standard error of the mean	1.03	0.35	0.13
Coefficient of Skewness	0.27 (*p* = 0.1563)	0.032 (*p* = 0.8640)	−0.53 (*p* = 0.0073)
Coefficient of Kurtosis	−0.72 (*p* = 0.0050)	−0.97 (*p* < 0.0001)	−0.52 (*p* = 0.0887)
Shapiro–Wilk test for normal distribution	W = 0.94 reject normality (*p* < 0.0001)	W = 0.96 reject normality (*p* = 0.0003)	W = 0.94 reject normality (*p* < 0.0001)

Abbreviations: DAPSA = Disease Activity for Psoriatic Arthritis; CPDAI = Composite Psoriatic Disease Activity Index; PASDAS = Psoriatic Arthritis Disease Activity Score; CI = confidence interval.

**Table 4 jpm-13-01153-t004:** Spearman correlation coefficients for PsA-5Ts and all other continuous composite indices of disease activity and patient-reported outcome (PRO) measures in the case study.

		CPDAI	PASDAS	PsA-5Ts	PSAID-12	SF-36
DAPSA	correlation coefficient *p*	0.720 <0.0001	0.829 <0.0001	0.664 <0.0001	0.423 <0.0001	−0.625 <0.0001
CPDAI	correlation coefficient *p*		0.766 <0.0001	0.582 <0.0001	0.402 <0.0001	−0.592 <0.0001
PASDAS	correlation coefficient *p*			0.694 <0.0001	0.529 <0.0001	−0.639 <0.0001
PsA-5Ts	correlation coefficient *p*				0.735 <0.0001	−0.469 <0.0001
PsAID-12	correlation coefficient *p*					−0.165 0.0389

Abbreviations: CPDAI = Composite Psoriatic Disease Activity Index; PASDAS = Psoriatic Arthritis Disease Activity Score; PsA-5Ts = Psoriatic Arthritis-5 Thermometer scales; DAPSA = Disease Activity for Psoriatic Arthritis; PsAID-12 = Psoriatic Arthritis Impact of Disease 12-item; SF-36 = Medical Outcomes Study Short-Form 36.

**Table 5 jpm-13-01153-t005:** Comparison of the area under the receiver operating characteristic curves for the probability of each index in identifying patients achieving or not achieving the Minimal Disease Activity (MDA).

Variable	AUC	SE ^a^	95% CI ^b^
DAPSA	0.963	0.014	0.920–0.986
CPDAI	0.924	0.021	0.871–0.961
PASDAS	0.955	0.016	0.909–0.981
PsA-5Ts	0.944	0.026	0.895–0.974
PsAID-12	0.917	0.024	0.863–0.955
SF-36	0.799	0.042	0.728–0.859

Abbreviations and legend: AUC = area under the receiver operating characteristic curve; SE = standard error; CI = confidence interval; DAPSA = Disease Activity for Psoriatic Arthritis; CPDAI = Composite Psoriatic Disease Activity Index; PASDAS = Psoriatic Arthritis Disease Activity Score; PsA-5Ts = Psoriatic Arthritis-5 Thermometer scales; PsAID-12 = Psoriatic Arthritis Impact of Disease 12-item; SF-36 = Medical Outcomes Study Short-Form 36; ^a^ = Hanley and McNeil 1982; ^b^ = Binomial exact.

## Data Availability

Data are available upon reasonable request to the corresponding author.

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
