# Peer review of "The Psoriatic Arthritis 5-Thermometer Scales (PsA-5Ts): Measurement Properties of a New Multidimensional Composite Tool for the Quick Assessment of the Overall Health Status in Psoriatic Arthritis"

_jpm, 2023, doi:10.3390/jpm13071153_

Round 1
Reviewer 1 Report
The Psoriatic Arthritis 5-Thermometer scales (PsA-5Ts): meas[1]urement properties of a new multidimensional composite tool for the quick assessment of the overall health status in Psoriatic Arthritis
This is an interesting study from Italy which introduces a new patient completed tool to assess health status in psoriatic arthritis. The authors describe the development of the tool, and perform some initial validation studies using real patient data. I have some comments:
1. More information is needed on the development stages of this instrument. What was the result of the literature search and the Delphi exercise? Were patients involved in any of the stages. In the item reduction stage were any patients involved? And how was the figure of 5 domains arrived at? It is uncanny that 4 of the 5 domains identified are the same as the top domains in the PsAID. How did the authors achieve the weighting?
2. Previous attempts at improving existing disease activity measures in PsA have included fatigue, as indicated by patient panels. However, these attempts have failed. I suspect the same would be true of depression.
3. I wonder if this instrument is less of a disease activity measure and more of an impact measure, like the PsAID. Hence the better correlation. The abstract conclusion implies that this new instrument is a better alternative to continuous scales, but these measure a different concept (disease activity) and so the comparison is invalid.
4. So it seems to me that the authors are introducing a measure of impact and it is the PsAID that they wish to replace – it is perhaps more relevant to focus on this measure for comparison. Would the PsAID work better if presented as a thermometer scale?
Author Response
Reviewer 1
The Psoriatic Arthritis 5-Thermometer scales (PsA-5Ts): meas[1][1]urement properties of a new multidimensional composite tool for the quick assessment of the overall health status in Psoriatic Arthritis
This is an interesting study from Italy which introduces a new patient completed tool to assess health status in psoriatic arthritis.
Thank you for the appreciation.
The authors describe the development of the tool, and perform some initial validation studies using real patient data. I have some comments:
- More information is needed on the development stages of this instrument. What was the result of the literature search and the Delphi exercise? Were patients involved in any of the stages. In the item reduction stage were any patients involved? And how was the figure of 5 domains arrived at? It is uncanny that 4 of the 5 domains identified are the same as the top domains in the PsAID. How did the authors achieve the weighting?
Thank you for your comments. We have tried to fill the highlighted methodological gaps by making extensive changes in the section on materials and methods. Accordingly, Table 1 has also been modified. In particular, the weight of the items was clarified and why the 5 items were retained; patients were involved only in a pretest but not in item generation. The fact that 4 of the 5 deemed items are shared with the PsAID-12 is not surprising, since the PsAID-12 was one of the starting tools of item generation.
- Previous attempts at improving existing disease activity measures in PsA have included fatigue, as indicated by patient panels. However, these attempts have failed. I suspect the same would be true of depression.
Thank you for the comment. Probably, due to the fact that the item generation was done by rheumatologists and not patients, fatigue and depression were considered as relevant.
- I wonder if this instrument is less of a disease activity measure and more of an impact measure, like the PsAID. Hence the better correlation. The abstract conclusion implies that this new instrument is a better alternative to continuous scales, but these measure a different concept (disease activity) and so the comparison is invalid.
Thank you for your comment. In order not to cause confusion, the last sentence in the abstract has been revised. PsA-5Ts is not a measure of disease activity but of overall health status.
- So it seems to me that the authors are introducing a measure of impact and it is the PsAID that they wish to replace – it is perhaps more relevant to focus on this measure for comparison. Would the PsAID work better if presented as a thermometer scale?
Thank you for the comment. Indeed there is some overlap with the PsAID-12, because of the issue mentioned above regarding item generation. On whether the PsAID-12 is better when presented as a thermometric scale, one cannot comment since it is presented as an NRS.
Reviewer 2 Report
Thank you for the opportunity this interesting manuscript regarding development of a clinimetrics tool for psoriatic arthritis, which may equip clinicians with improved tools to clinically evaluate PA with consistency and validity. Comments are generally related to presentation of material and findings to improve readability.
· Introduction – overall presents some interesting information but is long and somewhat difficult to follow - recommend to significantly shorten (i.e. 3 paragraphs) first introducing psoriatic arthritis, then the available composite measures and PROs for PA, then end with the difficulties / lack of standardized approaches for their application and the objective of the manuscript.
· Methods
o Recommend to further define each acronym, rely less on citations, and provide additional. Subheadings for each criteria evaluated.
o Define DAPSA, CPDAI and PASDAS when brought up again
· Discussion
o Recommend to revise first paragraph of discussion to not just state but interpret for the reader the significance of your findings and clinical application moving forward
o Move content as indicated from introduction to discussion, shortening introduction and revising discussion to remove additional background information on prevalence of PsA (reduce and move this to introduction) and simplify paragraph structure to center on PROs and how the findings of THIS study will advance what is currently available for PROs for PsA
o Include further reference to clinicometric data and science behind development of clinical measurements
Consider involving a native English speaker - minor editing may improve the manuscript's readability
Author Response
Reviewer 2
Comments and Suggestions for Authors
Thank you for the opportunity this interesting manuscript regarding development of a clinimetrics tool for psoriatic arthritis, which may equip clinicians with improved tools to clinically evaluate PA with consistency and validity.
Thank you for the appreciation.
Comments are generally related to presentation of material and findings to improve readability.
- Introduction – overall presents some interesting information but is long and somewhat difficult to follow - recommend to significantly shorten (i.e. 3 paragraphs) first introducing psoriatic arthritis, then the available composite measures and PROs for PA, then end with the difficulties / lack of standardized approaches for their application and the objective of the manuscript.
Thank you for your comments, the introduction has been amended as suggested.
- Methods
Recommend to further define each acronym, rely less on citations, and provide additional. Subheadings for each criteria evaluated.
Define DAPSA, CPDAI and PASDAS when brought up again
Changes made as requested, thank you.
- Discussion
Recommend to revise first paragraph of discussion to not just state but interpret for the reader the significance of your findings and clinical application moving forward
Move content as indicated from introduction to discussion, shortening introduction and revising discussion to remove additional background information on prevalence of PsA (reduce and move this to introduction) and simplify paragraph structure to center on PROs and how the findings of THIS study will advance what is currently available for PROs for PsA
Include further reference to clinicometric data and science behind development of clinical measurements
Thank you for your comments. The discussion has been amended as suggested.
Comments on the Quality of English Language
Consider involving a native English speaker - minor editing may improve the manuscript's readability
Some improvements of the English language have been made. Thank you
For the Editor:
Some bibliographic references have been updated, the overall number has been reduced as well as self-citations.
Round 2
Reviewer 1 Report
thank you for the additional material which clarifies the development of this tool
none